# Scalable DP-SGD: Shuffling vs. Poisson Subsampling

**Lynn Chua**
Google Research
chualynn@google.com

**Badih Ghazi**
Google Research
badihghazi@gmail.com

**Pritish Kamath**
Google Research
pritishk@google.com

**Ravi Kumar**
Google Research
ravi.k53@gmail.com

**Pasin Manurangsi**
Google Research
pasin@google.com

**Amer Sinha**
Google Research
amersinha@google.com

**Chiyuan Zhang**
Google Research
chiyuan@google.com

## Abstract

We provide new lower bounds on the privacy guarantee of the *multi-epoch* Adaptive Batch Linear Queries (ABLQ) mechanism with *shuffled batch sampling*, demonstrating substantial gaps when compared to *Poisson subsampling*; prior analysis was limited to a single epoch. Since the privacy analysis of Differentially Private Stochastic Gradient Descent (DP-SGD) is obtained by analyzing the ABLQ mechanism, this brings into serious question the common practice of implementing shuffling-based DP-SGD, but reporting privacy parameters as if Poisson subsampling was used. To understand the impact of this gap on the utility of trained machine learning models, we introduce a practical approach to implement Poisson subsampling *at scale* using massively parallel computation, and efficiently train models with the same. We compare the utility of models trained with Poisson-subsampling-based DP-SGD, and the optimistic estimates of utility when using shuffling, via our new lower bounds on the privacy guarantee of ABLQ with shuffling.

## 1 Introduction

A common approach for private training of differentiable models, such as neural networks, is to apply first-order methods with noisy gradients. This general framework is known as DP-SGD (Differentially Private Stochastic Gradient Descent) [Abadi et al., 2016]; the framework itself is compatible with any optimization sub-routine. Multiple open source implementations exist for applying DP-SGD in practice, namely, Tensorflow Privacy, JAX Privacy [Balle et al., 2022] and PyTorch Opacus [Yousefpour et al., 2021]; and DP-SGD has been applied widely in various machine learning domains [e.g., Tramer and Boneh, 2020, De et al., 2022, Bu et al., 2022, Chen et al., 2020, Dockhorn et al., 2023, Anil et al., 2022, He et al., 2022, Igamberdiev et al., 2024, Tang et al., 2024].

DP-SGD (Algorithm 1) processes the training data in a sequence of steps, where at each step, a noisy estimate of the average gradient over a mini-batch is computed and used to perform a first-order update over the differentiable model. To obtain the noisy (average) gradient, the gradient $g$ for each example in the mini-batch is *clipped* to have norm at most $C$ (a pre-determined fixed bound), by setting $[g]_C := g \cdot \min\{1, C/\|g\|_2\}$, and computing the sum over the batch; then independent zero-mean noise drawn from the Gaussian distribution of scale $\sigma C$ is added to each coordinate of the summed gradient. This could then be scaled by the "target" mini-batch size to obtain a noisy

38th Conference on Neural Information Processing Systems (NeurIPS 2024).

---

**Algorithm 1** DP-SGD: Differentially Private Stochastic Gradient Descent [Abadi et al., 2016]

---

**Params:** Batch sampler $\mathcal{B}$ (samples $T$ batches, with "target" batch size $b$), differentiable loss
 $\ell : \mathbb{R}^d \times \mathcal{X} \to \mathbb{R}^d$, initial model state $w_0$, clipping norm $C$, noise scale $\sigma$.
**Input:** Dataset $\boldsymbol{x} = (x_1, \ldots, x_n)$.
**Output:** Final model state $\boldsymbol{w}_T \in \mathbb{R}^d$.
 $(S_1, \ldots, S_T) \leftarrow \mathcal{B}(n)$
 **for** $t = 1, \ldots, T$ **do**
  $g_t \leftarrow \frac{1}{b} \left( \mathcal{N}(0, \sigma^2 C^2 I_d) + \sum_{x \in S_t} [\nabla_{\boldsymbol{w}} \ell(\boldsymbol{w}; x)]_C \right)$
  $\boldsymbol{w}_t \leftarrow \boldsymbol{w}_{t-1} - \eta_t g_t$       $\triangleright$ Could also be some other optimization method.
 **return** $w_T$

---

average gradient.[1] The privacy guarantee of the mechanism depends on the following parameters: the noise scale $\sigma$, the number of examples in the training dataset, the size of mini-batches, the number of training steps, and the *mini-batch generation process*.

In practice, almost all deep learning systems generate mini-batches of fixed-size by sequentially going over the dataset, possibly applying a global *shuffling* of all the examples in the dataset for each training epoch; each epoch corresponds to a single pass over the dataset, and the ordering of the examples may be kept the same or resampled between different epochs. However, performing the privacy analysis for such a mechanism has appeared to be technically difficult due to correlation between the different mini-batches. Abadi et al. [2016] instead consider a different mini-batch generation process of *Poisson subsampling*, wherein each mini-batch is generated independently by including each example with a fixed probability. This mini-batch generation process is however rarely implemented in practice, and consequently it has become common practice to use some form of shuffling in applications, but to report privacy parameters as if Poisson subsampling was used (see, e.g., the survey by Ponomareva et al. [2023, Section 4.3]). A notable exception is the PyTorch Opacus library [Yousefpour et al., 2021] that supports the option of Poisson subsampling; however, this implementation only works well for datasets that allow efficient random access (for instance by loading it entirely into memory). To the best of knowledge, Poisson subsampling has not been used for training with DP-SGD on massive datasets.

The privacy analysis of DP-SGD is usually performed by viewing it as a post-processing of an *Adaptive Batch Linear Queries (*ABLQ*)* mechanism that releases the estimates of a sequence of adaptively chosen linear queries on the mini-batches (formal definitions in Section 2.1). Chua et al. [2024] showed that the privacy loss of ABLQ with shuffling can be significantly higher than that with Poisson subsampling for small values of $\sigma$. Even though their analysis only applied to a *single epoch* mechanism, this has put under serious question the aforementioned common practice of implementing DP-SGD with some form of shuffling while reporting privacy parameters assuming Poisson subsampling. The motivating question for our work is:

> *Which batch sampler provides the best utility for models trained with* DP-SGD,
> *when applied with the correct corresponding privacy accounting?*

## 1.1 Contributions

Our contributions are summarized as follows.

**Privacy Analysis of Multi-Epoch ABLQ with Shuffling.** We provide *lower bounds* on the privacy guarantees of shuffling-based ABLQ to handle *multiple epochs*. We consider the cases of both (i) *Persistent Shuffling*, wherein the examples are globally shuffled once and the order is kept the same between epochs, and (ii) *Dynamic Shuffling*, wherein the examples are globally shuffled independently for each epoch. Since our technique provides a lower bound on the privacy guarantee, the utility of the models obtained via shuffling-based DP-SGD with this privacy accounting is an optimistic estimate of the utility under the correct accounting.

**Scalable Implementation of DP-SGD with Poisson Subsampling via Truncation.** Variable batches are typically inconvenient to handle in deep learning systems. For example, upon a change in the input shape, `jax.jit` triggers a recompilation of the computation graph, and `tf.function` will

---

[1]As explained later, for Poisson batch sampler, the mini-batch size is not a constant, but the scaling has to be done with the "target" mini-batch size, and not the realized mini-batch size.

retrace the computation graph. Additionally, Google TPUs require all operations to have fixed input and output shapes. We introduce *truncated Poisson subsampling* to circumvent variable batch sizes. In particular, we choose an upper bound on the maximum batch size $B$ that our training can handle, and given any variable size batch $b$, if $b \geq B$, we randomly sub-select $B$ examples to retain in the batch, and if $b < B$, we pad the batch with $B - b$ dummy examples *with zero weight*. This deviates slightly from the standard Poisson subsampling process since our batch sizes can never exceed $B$. We choose $B$ to be sufficiently larger than the expected batch size, so that the probability that the sampled batch size $b$ exceeds the maximum allowed batch size $B$ is small. We provide a modification to the analysis of ABLQ with Poisson subsampling in order to handle this difference.

Generating these truncated Poisson subsampled batches can be difficult when the dataset is too large to fit in memory. We provide a scalable approach to the generation of batches with truncated Poisson subsampling using *massively parallel computation* [Dean and Ghemawat, 2004]. This can be easily specified using frameworks like beam [Apache Beam] and implemented on distributed platforms such as Apache Flink, Apache Spark, or Google Cloud Dataflow.

Our detailed experimental results are presented in Section 4, and summarized below:

- DP-SGD with Shuffle batch samplers performs similarly to Poisson subsampling for the same $\sigma$.
- However, DP-SGD with Shuffle batch samplers, with our optimistic privacy accounting, perform worse than Poisson subsampling in high privacy regimes (small values of $\varepsilon$).

Thus, our results suggest that Poisson subsampling is a viable option for implementing DP-SGD at scale, with almost no loss in utility compared to the traditional approach that uses shuffling with (incorrect) accounting assuming Poisson subsampling.

## 1.2 Related Work

Chua et al. [2024] demonstrated gaps in the privacy analysis of ABLQ using shuffling and Poisson subsampling, by providing a *lower bound* on the privacy guarantee of ABLQ with shuffling; their technique, however, was specialized for one epoch. We extend their technique to the *multi-epoch* version of ABLQ with shuffling and provide lower bounds for both persistent and dynamic batching.

Lebeda et al. [2024] also point out gaps in the privacy analysis of ABLQ with Poisson subsampling and with sampling batches of fixed size independently, showing that the latter has worse privacy guarantees than Poisson subsampling. We do not cover this sampling in our experimental study, since sampling independent batches of fixed size is not commonly implemented in practice, and DP-SGD using this sampling is only expected to be worse as compared to Poisson subsampling.

Yousefpour et al. [2021] report the model utility (and computational cost overhead) under training with DP-SGD with Poisson subsampling. However, to the best of our knowledge, there is no prior work that has compared the model utility of DP-SGD under Poisson subsampling with that under shuffling, let alone compared it against DP-SGD under (Dynamic/Persistent) shuffling or studied the gaps between the privacy accounting of the two approaches.

One possible gap between the privacy analysis of DP-SGD and ABLQ is that the former only releases the final iterate, whereas the latter releases the responses to all the queries. An interesting result by Annamalai [2024] shows that in general the privacy analysis of the *last-iterate* of DP-SGD cannot be improved over that of ABLQ, when using Poisson subsampling. This suggests that at least without any further assumptions, e.g., on the loss function, it is not possible to improve the privacy analysis of DP-SGD beyond that provided by ABLQ; this is in contrast to the techniques of privacy amplification by iteration for convex loss functions [e.g. Feldman et al., 2018, Altschuler and Talwar, 2022].

## 2 Preliminaries

A differentially private (DP) mechanism $\mathcal{M} : \mathcal{X}^* \to \Delta_{\mathcal{O}}$ can be viewed as a mapping from input datasets to distributions over an output space, namely, on input *dataset* $\boldsymbol{x} = (x_1, \ldots, x_n)$ where each *example* $x_i \in \mathcal{X}$, $\mathcal{M}(\boldsymbol{x})$ is a probability measure over the output space $\mathcal{O}$; for ease of notation, we often refer to the corresponding random variable also as $\mathcal{M}(\boldsymbol{x})$. Two datasets $\boldsymbol{x}$ and $\boldsymbol{x}'$ are said to be *adjacent*, denoted $\boldsymbol{x} \sim \boldsymbol{x}'$, if they "differ in one example"; in particular, we use the "zeroing-out" adjacency defined shortly.

---

**Algorithm 2** ABLQ$_\mathcal{B}$: Adaptive Batch Linear Queries (as formalized in Chua et al. [2024])

---

**Params:** Batch sampler $\mathcal{B}$ (samples $T$ batches), noise scale $\sigma$, and (adaptive) query method $\mathcal{A}$ : $(\mathbb{R}^d)^* \times \mathcal{X} \to \mathbb{B}^d$.
**Input:** Dataset $\boldsymbol{x} = (x_1, \ldots, x_n)$.
**Output:** Query estimates $g_1, \ldots, g_T \in \mathbb{R}^d$
  $(S_1, \ldots, S_T) \leftarrow \mathcal{B}(n)$
  **for** $t = 1, \ldots, T$ **do**
    $\psi_t(\cdot) := \mathcal{A}(g_1, \ldots, g_{t-1}; \cdot)$
    $g_t \leftarrow e_t + \sum_{i \in S_t} \psi_t(x_i)$ for $e_t \sim \mathcal{N}(0, \sigma^2 I_d)$
  **return** $(g_1, \ldots, g_T)$

---

| | |
|---|---|
| **Algorithm 3** $\Pi_{b,T}(n; \vec{\pi})$: Permutation Batch Sampler | **Algorithm 4** $\mathcal{P}_{b,B,T}$: Truncated Poisson Batch Sampler |

**Algorithm 3** $\Pi_{b,T}(n; \vec{\pi})$: Permutation Batch Sampler

**Params:** Batch size $b$, number of batches $T$.
**Input:** Number of examples $n$ s.t. $E := bT/n$ is an integer (number of epochs), $S := n/b$ is an integer (number of steps per epoch); a list $\vec{\pi} = \pi_0, \ldots, \pi_{E-1}$, where each $\pi_e$ is a permutation of $[n]$.
**Output:** Seq. $S_1, \ldots, S_T \subseteq [n]$ of batches.
  **for** $e = 0, \ldots, E - 1$ **do**
    **for** $s = 0, \ldots, S - 1$ **do**
      $t = e \cdot S + s + 1$
      $S_t \leftarrow \{\pi_e(sb+1), \ldots, \pi_e(sb+b)\}$
  **return** $S_1, \ldots, S_T$

**Algorithm 4** $\mathcal{P}_{b,B,T}$: Truncated Poisson Batch Sampler

**Params:** Target batch size $b$, max batch size $B$, number of batches $T$.
**Input:** Number of examples $n$.
**Output:** Seq. $S_1, \ldots, S_T \subseteq [n]$ of batches, with $|S_t| \leq B$.
  **for** $t = 1, \ldots, T$ **do**
    $S_t \leftarrow \emptyset$
    **for** $i = 1, \ldots, n$ **do**
      $S_t \leftarrow \begin{cases} S_t \cup \{i\} & \text{with prob. } b/n \\ S_t & \text{with prob. } 1 - b/n \end{cases}$
    **if** $|S_t| > B$ **then**
      $S_t \leftarrow$ arbitrary subset of $S_t$ of size $B$
  **return** $S_1, \ldots, S_T$

---

**Definition 2.1** (DP). For $\varepsilon, \delta \geq 0$, a mechanism $\mathcal{M}$ satisfies $(\varepsilon, \delta)$-DP if for all "adjacent" datasets $\boldsymbol{x} \sim \boldsymbol{x}'$, and for any (measurable) event $\Gamma$ it holds that $\Pr[\mathcal{M}(\boldsymbol{x}) \in \Gamma] \leq e^\varepsilon \Pr[\mathcal{M}(\boldsymbol{x}') \in \Gamma] + \delta$.

For any mechanism $\mathcal{M}$, we use $\delta_\mathcal{M} : \mathbb{R}_{\geq 0} \to [0, 1]$ to denote its *privacy loss curve*, namely $\delta_\mathcal{M}(\varepsilon)$ is the smallest $\delta$ such that $\mathcal{M}$ satisfies $(\varepsilon, \delta)$-DP; $\varepsilon_\mathcal{M} : [0, 1] \to \mathbb{R}_{\geq 0}$ is defined similarly.

## 2.1 Adaptive Batch Linear Queries Mechanism

Following the notation in Chua et al. [2024], we study the adaptive batch linear queries mechanism ABLQ$_\mathcal{B}$ (Algorithm 2) using a *batch sampler* $\mathcal{B}$ and an *adaptive query method* $\mathcal{A}$, defined. The batch sampler $\mathcal{B}$ can be any algorithm that randomly samples a sequence $S_1, \ldots, S_T$ of batches. ABLQ$_\mathcal{B}$ operates by processing the batches in a sequential order, and produces a sequence $(g_1, \ldots, g_T)$, where the response $g_t \in \mathbb{R}^d$ is produced as the sum of $\psi_t(x)$ over the batch $S_t$ with added zero-mean Gaussian noise of scale $\sigma$ to all coordinates, where the query $\psi_t : \mathcal{X} \to \mathbb{B}^d$ (for $\mathbb{B}^d := \{v \in \mathbb{R}^d : \|v\|_2 \leq 1\}$) is produced by the adaptive query method $\mathcal{A}$, based on the previous responses $g_1, \ldots, g_{t-1}$. DP-SGD can be viewed as a post-processing of an adaptive query method that maps examples to the clipped gradient at the last iterate, namely $\psi_t(x) := [\nabla_{\boldsymbol{w}} \ell(\boldsymbol{w}_{t-1}, x)]_1$ (we treat the clipping norm $C = 1$ for simplicity, as it is just a scaling term).

In this work, we consider the following *multi-epoch* batch samplers: Deterministic $\mathcal{D}_{b,T}$, Persistent Shuffle $\mathcal{S}^\diamond_{b,T}$, and Dynamic Shuffle $\mathcal{S}^\circlearrowleft_{b,T}$ batch sampler defined as instantiations of Algorithm 3 in Figure 1 and truncated Poisson $\mathcal{P}_{b,B,T}$ (Algorithm 4); we drop the subscripts of each sampler whenever it is clear from context. Note that, while $\mathcal{P}_{b,B,T}$ has no restriction on the value of $n$, the samplers $\mathcal{D}_{b,T}$, $\mathcal{S}^\diamond_{b,T}$, and $\mathcal{S}^\circlearrowleft_{b,T}$ require that the number of examples $n$ is such that $E := bT/n$ and $S := n/b$ are integers, where $E$ corresponds to the number of epochs and $S$ corresponds to the number of steps per epoch. We call the tuple $(n, b, T)$ as "valid" if that holds, and we will often implicitly assume that this holds. Also note that $\mathcal{P}_{b,B,T}$ corresponds to the standard Poisson subsampling without truncation when $B = \infty$. We use $\delta_\mathcal{B}(\varepsilon)$ to denote the *privacy loss curve* of ABLQ$_\mathcal{B}$ for any $\mathcal{B} \in \{\mathcal{D}, \mathcal{P}, \mathcal{S}^\diamond, \mathcal{S}^\circlearrowleft\}$, where other parameters such as $\sigma, T$, etc. are implicit. Namely, for all $\varepsilon > 0$, let $\delta_\mathcal{B}(\varepsilon)$ be the smallest $\delta \geq 0$ such that ABLQ$_\mathcal{B}$ satisfies $(\varepsilon, \delta)$-DP for *all* choices of the underlying adaptive query method $\mathcal{A}$. We define $\varepsilon_\mathcal{B}(\delta)$ similarly. Finally, we define $\sigma_\mathcal{B}(\varepsilon, \delta)$ as the smallest $\sigma$ such that ABLQ$_\mathcal{B}$ satisfies $(\varepsilon, \delta)$-DP, with other parameters being implicit in $\mathcal{B}$.

> **Deterministic Batch Sampler $\mathcal{D}_{b,T}(n)$:** Realized as $\Pi_{b,T}(n; \text{Id}, \text{Id}, \ldots)$. where $\text{Id}$ is the identity permutation, i.e., the data is not permuted.
>
> **Persistent Shuffle Batch Sampler $\mathcal{S}^{\diamond}_{b,T}$:** Realized as $\Pi_{b,T}(n; \pi, \pi, \ldots)$, where $\pi$ is a random permutation over $[n]$, i.e., the data is shuffled once and the order is persistent across epochs.
>
> **Dynamic Shuffle Batch Sampler $\mathcal{S}^{\circlearrowright}_{b,T}$:** Realized as $\Pi_{b,T}(n; \pi_0, \pi_1, \ldots)$, where $\pi_e$'s are i.i.d. random permutations over $[n]$, i.e., the data is reshuffled in each epoch.

Figure 1: Various natural instantiations of the permutation batch sampler.

**Adjacency notion.** The common notion of *Add-Remove* adjacency is not applicable for mechanisms such as $\text{ABLQ}_{\mathcal{D}}$, $\text{ABLQ}_{\mathcal{S}^{\diamond}}$, $\text{ABLQ}_{\mathcal{S}^{\circlearrowright}}$ because these methods require that $bT/n$ and $n/b$ are integers, and changing $n$ by $\pm 1$ does not respect this requirement. And while the other common notion of *Substitution* is applicable for all the mechanisms we consider, the standard analysis for $\text{ABLQ}_{\mathcal{P}}$ is done w.r.t Add-Remove adjacency [Abadi et al., 2016, Mironov, 2017]. Therefore, we use the "Zeroing-out" adjacency introduced by Kairouz et al. [2021], namely we consider the augmented input space $\mathcal{X}_{\perp} := \mathcal{X} \cup \{\perp\}$ where any adaptive query method $\mathcal{A}$ is extended as $\mathcal{A}(g_1, \ldots, g_t; \perp) := \mathbf{0}$ for all $g_1, \ldots, g_t \in \mathbb{R}^d$. Datasets $\boldsymbol{x}, \boldsymbol{x}' \in \mathcal{X}_{\perp}^n$ are said to be *zero-out* adjacent if there exists $i$ such that $\boldsymbol{x}_{-i} = \boldsymbol{x}'_{-i}$, and exactly one of $\{x_i, x'_i\}$ is in $\mathcal{X}$ and the other is $\perp$. We use $\boldsymbol{x} \to_z \boldsymbol{x}'$ to specifically denote adjacent datasets with $x_i \in \mathcal{X}$ and $x'_i = \perp$. Thus $\boldsymbol{x} \sim \boldsymbol{x}'$ if either $\boldsymbol{x} \to_z \boldsymbol{x}'$ or $\boldsymbol{x}' \to_z \boldsymbol{x}$.

## 2.2 Dominating Pairs

For two probability density functions $P$ and $Q$ and $\alpha, \beta \in \mathbb{R}_{\geq 0}$, we use $\alpha P + \beta Q$ to denote the weighted sum of the density functions. We use $P \otimes Q$ to denote the product distribution sampled as $(\omega_1, \omega_2)$ for $\omega_1 \sim P, \omega_2 \sim Q$, and, $P^{\otimes T}$ to denote the $T$-fold product distribution $P \otimes \cdots \otimes P$. For all $\varepsilon \in \mathbb{R}$, the $e^{\varepsilon}$-*hockey stick divergence* between $P$ and $Q$ is $D_{e^{\varepsilon}}(P \| Q) := \sup_{\Gamma} P(\Gamma) - e^{\varepsilon} Q(\Gamma)$. Thus, by definition a mechanism $\mathcal{M}$ satisfies $(\varepsilon, \delta)$-DP iff for all adjacent $\boldsymbol{x} \sim \boldsymbol{x}'$, it holds that $D_{e^{\varepsilon}}(\mathcal{M}(\boldsymbol{x}) \| \mathcal{M}(\boldsymbol{x}')) \leq \delta$.

**Definition 2.2** (Dominating Pair [Zhu et al., 2022]). *The pair $(P, Q)$ dominates the pair $(A, B)$ (denoted $(P, Q) \succcurlyeq (A, B)$) if $D_{e^{\varepsilon}}(P \| Q) \geq D_{e^{\varepsilon}}(A \| B)$ holds for all $\varepsilon \in \mathbb{R}$. We say that $(P, Q)$ dominates a mechanism $\mathcal{M}$ (denoted $(P, Q) \succcurlyeq \mathcal{M}$) if $(P, Q) \succcurlyeq (\mathcal{M}(\boldsymbol{x}), \mathcal{M}(\boldsymbol{x}'))$ for all adjacent $\boldsymbol{x} \to_z \boldsymbol{x}'$.*

If $(P, Q) \succcurlyeq \mathcal{M}$, then for all $\varepsilon \geq 0$, it holds that $\delta_{\mathcal{M}}(\varepsilon) \leq \max\{D_{e^{\varepsilon}}(P \| Q), D_{e^{\varepsilon}}(Q \| P)\}$, and conversely, if there exists adjacent datasets $\boldsymbol{x} \to_z \boldsymbol{x}'$ such that $(\mathcal{M}(\boldsymbol{x}), \mathcal{M}(\boldsymbol{x}')) \succcurlyeq (P, Q)$, then $\delta_{\mathcal{M}}(\varepsilon) \geq \max\{D_{e^{\varepsilon}}(P \| Q), D_{e^{\varepsilon}}(Q \| P)\}$. When both of these hold, we say that $(P, Q)$ *tightly dominates* the mechanism $\mathcal{M}$ (denoted $(P, Q) \equiv \mathcal{M}$) and in this case it holds that $\delta_{\mathcal{M}}(\varepsilon) = \max\{D_{e^{\varepsilon}}(P \| Q), D_{e^{\varepsilon}}(Q \| P)\}$. Thus, tightly dominating pairs completely characterize the privacy loss of a mechanism (although they are not guaranteed to exist for all mechanisms). Dominating pairs behave nicely under mechanism compositions: if $(P_1, Q_1) \succcurlyeq \mathcal{M}_1$ and $(P_2, Q_2) \succcurlyeq \mathcal{M}_2$, then $(P_1 \otimes P_2, Q_1 \otimes Q_2) \succcurlyeq \mathcal{M}_1 \circ \mathcal{M}_2$, where $\mathcal{M}_1 \circ \mathcal{M}_2$ denotes the (adaptively) composed mechanism.

## 3 Privacy analysis of *multi-epoch* $\text{ABLQ}_{\mathcal{B}}$

We discuss the privacy analysis of $\text{ABLQ}_{\mathcal{B}}$ for $\mathcal{B} \in \{\mathcal{D}, \mathcal{P}, \mathcal{S}^{\diamond}, \mathcal{S}^{\circlearrowright}\}$ via dominating pairs.

**Privacy analysis for $\text{ABLQ}_{\mathcal{D}}$.** A single epoch of the $\text{ABLQ}_{\mathcal{D}}$ mechanism corresponds to a Gaussian mechanism with noise scale $\sigma$. And thus, $E := bT/n$ epochs of the $\text{ABLQ}_{\mathcal{D}}$ mechanism corresponds to an $E$-fold composition of the Gaussian mechanism, which is privacy-wise equivalent to a Gaussian mechanism with noise scale $\sigma/\sqrt{E}$ [Dong et al., 2019, Corollary 3.3]. Thus, a closed-form expression for $\delta_{\mathcal{D}}(\varepsilon)$ exists via the dominating pair $(P_{\mathcal{D}} = \mathcal{N}(1, \frac{\sigma^2}{E}), Q_{\mathcal{D}} = \mathcal{N}(0, \frac{\sigma^2}{E}))$.

**Theorem 3.1** (Balle and Wang [2018, Theorem 8]). *For all $\sigma > 0$, $\varepsilon \geq 0$, and valid $n$, $b$, $T$, it holds that*

$$\delta_{\mathcal{D}}(\varepsilon) = \Phi\left(-\sigma'\varepsilon + \tfrac{1}{2\sigma'}\right) - e^{\varepsilon}\Phi\left(-\sigma'\varepsilon - \tfrac{1}{2\sigma'}\right), \qquad \text{where } \sigma' = \tfrac{\sigma}{\sqrt{E}},$$

*and $\Phi(\cdot)$ is the cumulative density function (CDF) of the standard normal random variable $\mathcal{N}(0, 1)$.*

**Privacy analysis of $\mathsf{ABLQ}_{\mathcal{P}}$.** First, let us consider the case of Poisson subsampling without truncation, namely $B = \infty$. Zhu et al. [2022] showed[2] that the tightly dominating pair for a single step of $\mathsf{ABLQ}_{\mathcal{P}}$, a Poisson sub-sampled Gaussian mechanism, is given by the pair $(U = (1-q)\mathcal{N}(0, \sigma^2) + q\mathcal{N}(1, \sigma^2), V = \mathcal{N}(0, \sigma^2))$, where $q$ is the sub-sampling probability of each example, namely $q = b/n$. Since $\mathsf{ABLQ}_{\mathcal{P}}$ is a $T$-fold composition of this Poisson subsampled Gaussian mechanism, it follows that $(P_{\mathcal{P}} := U^{\otimes T}, Q_{\mathcal{P}} := V^{\otimes T}) \equiv \mathsf{ABLQ}_{\mathcal{P}}$.

A finite value of $B$ however changes the mechanism slightly. In order to handle this, we use the following proposition, where $d_{\mathrm{TV}}(P, P')$ denotes the *statistical distance* between $P$ and $P'$.

**Proposition 3.2.** *For distributions $P, P', Q, Q'$ such that $d_{\mathrm{TV}}(P, P'), d_{\mathrm{TV}}(Q, Q') \leq \eta$, and $D_{e^\varepsilon}(P'\|Q') \leq \delta$, then $D_{e^\varepsilon}(P\|Q) \leq \delta + \eta(1 + e^\varepsilon)$.*

*Proof.* For any event $\Gamma$ we have that

$$P(\Gamma) \overset{(i)}{\leq} P'(\Gamma) + \eta \overset{(ii)}{\leq} e^\varepsilon Q'(\Gamma) + \delta + \eta \overset{(iii)}{\leq} e^\varepsilon(Q(\Gamma) + \eta) + \delta + \eta = e^\varepsilon Q(\Gamma) + \delta + \eta(1 + e^\varepsilon),$$

where (i) follows from $d_{\mathrm{TV}}(P, P') \leq \eta$, (ii) follows from $D_{e^\varepsilon}(P'\|Q') \leq \delta$ and (iii) follows from $d_{\mathrm{TV}}(Q, Q') \leq \eta$. Thus, we get that $D_{e^\varepsilon}(P\|Q) \leq \delta + \eta(1 + e^\varepsilon)$. $\qquad\square$

The batch size $|S_t|$ before truncation in $\mathcal{P}_{b,B,T}$ is distributed as the binomial distribution $\mathrm{Bin}(n, b/n)$, and thus, by a union bound over the events that the sampled batch size $|S_t| > B$ at any step, it follows that for any input dataset $\boldsymbol{x}$,

$$d_{\mathrm{TV}}(\mathsf{ABLQ}_{\mathcal{P}_{b,B,T}}(\boldsymbol{x}), \mathsf{ABLQ}_{\mathcal{P}_{b,\infty,T}}(\boldsymbol{x})) \leq T \cdot \Psi(n, b, B),$$

where $\Psi(n, b, B) := \Pr_{r \sim \mathrm{Bin}(n,b/n)}[r > B]$. Applying Proposition 3.2 we get

**Theorem 3.3.** *For all $\sigma > 0$, $\varepsilon \geq 0$, and integers $b$, $n \geq b$, $B \geq b$, $T$, it holds that*

$$\delta_{\mathcal{P}}(\varepsilon) \leq \max\{D_{e^\varepsilon}(P_{\mathcal{P}}\|Q_{\mathcal{P}}), D_{e^\varepsilon}(Q_{\mathcal{P}}\|P_{\mathcal{P}})\} + T \cdot (1 + e^\varepsilon) \cdot \Psi(n, b, B).$$

While the hockey stick divergences $D_{e^\varepsilon}(P_{\mathcal{P}}\|Q_{\mathcal{P}})$ and $D_{e^\varepsilon}(Q_{\mathcal{P}}\|P_{\mathcal{P}})$ do not have closed-form expressions, upper bounds on these can be obtained using privacy accountants based on the methods of Rényi DP (RDP) [Mironov, 2017] and *privacy loss distributions (PLD)* [Meiser and Mohammadi, 2018, Sommer et al., 2019]; the latter admits numerically accurate algorithms [Koskela et al., 2020, Gopi et al., 2021, Ghazi et al., 2022, Doroshenko et al., 2022], with multiple open-source implementations [Prediger and Koskela, 2020, Google's DP Library., 2020, Microsoft., 2021].

Note that $\Psi(n, b, B)$ can be made arbitrarily small by increasing $B$, which affects the computation cost. In particular, given a target privacy parameter $(\varepsilon, \delta)$, we can, for example, work backwards to first choose $B$ such that $\Psi(n, b, B) \cdot T \cdot (1 + e^\varepsilon) \leq 10^{-5} \cdot \delta$, and then choose the noise scale $\sigma$ such that $\max\{D_{e^\varepsilon}(P_{\mathcal{P}}\|Q_{\mathcal{P}}), D_{e^\varepsilon}(Q_{\mathcal{P}}\|P_{\mathcal{P}})\} \leq (1 - 10^{-5}) \cdot \delta$, using aforementioned privacy accounting libraries. Notice that our use of Proposition 3.2 is likely not the optimal approach to account for the batch truncation. We do not optimize this further because we find that this approach already provides very minimal degradation to the choice of $\sigma$ for a modest value of $B$ relative to $b$. A more careful analysis could at best result in a slightly smaller $B$, which we do not consider as significant; see Figures 3 and 4 for more details.

**Privacy analysis of $\mathsf{ABLQ}_{\mathcal{S}\diamond}$.** Obtaining the exact privacy guarantee for $\mathsf{ABLQ}_{\mathcal{S}}$ has been an open problem in the literature. Our starting point is the approach introduced by Chua et al. [2024] to prove a lower bound in the single epoch setting. Let the input space be $\mathcal{X} = [-1, 1]$, the (non-adaptive) query method $\mathcal{A}$ that produces the query $\psi_t(x) = x$, and consider the adjacent datasets:

$$\boldsymbol{x} = (x_1 = -1, \ldots, x_{n-1} = -1, x_n = 1) \quad \text{and} \quad \boldsymbol{x}' = (x_1 = -1, \ldots, x_{n-1} = -1, x_n = \perp).$$

Recall that the number of epochs is $E := bT/n$ and the number of steps per epoch is $S := n/b$. By considering the same setting, it is easy to see that the distributions $U = \mathsf{ABLQ}_{\mathcal{S}\diamond}(\boldsymbol{x})$ and $V = \mathsf{ABLQ}_{\mathcal{S}\diamond}(\boldsymbol{x}')$ are given as:

$$U = \sum_{s=1}^{S} \frac{1}{S} \cdot \mathcal{N}(-b \cdot \boldsymbol{1} + 2f_s, \sigma^2 I_T), \quad \text{and} \quad V = \sum_{s=1}^{S} \frac{1}{S} \cdot \mathcal{N}(-b \cdot \boldsymbol{1} + f_s, \sigma^2 I_T),$$

---

[2]Also implicit in prior work [Koskela et al., 2020].

where $f_s \in \mathbb{R}^T$ is the sum of basis vectors $\sum_{\ell=0}^{E-1} e_{\ell S+s}$, and $\mathbf{1}$ denotes the all-1's vector in $\mathbb{R}^T$. Basically, $f_s$ is the indicator vector encoding the batches that the differing example gets assigned to; in persistent shuffling, an example gets assigned to the $s$th batch within each epoch for a random $s \in \{1, \ldots, S\}$. Shifting the distributions by $b \cdot \mathbf{1}$ and projecting to the span of $\{f_s : s \in [S]\}$ does not change the hockey stick divergence $D_{e^\varepsilon}(U\|V)$, hence we might as well consider the pair

$$U' := \sum_{s=1}^{S} \tfrac{1}{S} \cdot \mathcal{N}(2\sqrt{E}e_s, \sigma^2 I_S) \quad \text{and} \quad V' := \sum_{s=1}^{S} \tfrac{1}{S} \cdot \mathcal{N}(\sqrt{E}e_s, \sigma^2 I_S).$$

By scaling down the distributions by $\sqrt{E}$ on all coordinates we arrive at the following pair:

$$P_{\mathcal{S}^\diamond} := \sum_{s=1}^{S} \tfrac{1}{S} \cdot \mathcal{N}(2e_s, \tfrac{\sigma^2}{E}I) \quad \text{and} \quad Q_{\mathcal{S}^\diamond} := \sum_{s=1}^{S} \tfrac{1}{S} \cdot \mathcal{N}(e_s, \tfrac{\sigma^2}{E}I). \tag{1}$$

The pair $(P_{\mathcal{S}^\diamond}, Q_{\mathcal{S}^\diamond})$ is essentially same as the pair obtained by Chua et al. [2024], with $\sigma$ replaced by $\sigma/\sqrt{E}$, and we get the following:

**Proposition 3.4.** *For all $\sigma > 0$, $\varepsilon \geq 0$ and all valid $n$, $b$, $T$, it holds that*

$$\delta_{\mathcal{S}^\diamond}(\varepsilon) \geq \max\{D_{e^\varepsilon}(P_{\mathcal{S}^\diamond}\|Q_{\mathcal{S}^\diamond}), D_{e^\varepsilon}(Q_{\mathcal{S}^\diamond}\|P_{\mathcal{S}^\diamond})\}.$$

Following Chua et al. [2024], we can obtain a lower bound as $\delta_{\mathcal{S}^\diamond}(\varepsilon) \geq P_{\mathcal{S}^\diamond}(\Gamma) - e^\varepsilon Q_{\mathcal{S}^\diamond}(\Gamma)$ for any $\Gamma$, and in particular, we consider events of the form $\Gamma_C := \{w \in \mathbb{R}^S : \max_s w_s > C\}$ for various values of $C$. $P_{\mathcal{S}^\diamond}(\Gamma_C)$ and $Q_{\mathcal{S}^\diamond}(\Gamma_C)$ are efficient to compute as

$$P_{\mathcal{S}^\diamond}(\Gamma_C) = 1 - \Phi\left(\tfrac{C-2}{\sigma/\sqrt{E}}\right) \cdot \Phi\left(\tfrac{C}{\sigma/\sqrt{E}}\right)^{S-1} \quad \text{and} \quad Q_{\mathcal{S}^\diamond}(\Gamma_C) = 1 - \Phi\left(\tfrac{C-1}{\sigma/\sqrt{E}}\right) \cdot \Phi\left(\tfrac{C}{\sigma/\sqrt{E}}\right)^{S-1}.$$

Thus, using Proposition 3.4, we get that

**Theorem 3.5.** *For all $\sigma > 0$, $\varepsilon \geq 0$, and all valid $n$, $b$, $T$, it holds that*

$$\delta_{\mathcal{S}^\diamond}(\varepsilon) \geq \sup_{C \in \mathbb{R}} P_{\mathcal{S}^\diamond}(\Gamma_C) - Q_{\mathcal{S}^\diamond}(\Gamma_C).$$

**Privacy analysis of ABLQ$_{\mathcal{S}^\circ}$.** Our starting point for providing a lower bound on $\delta_{\mathcal{S}^\circ}(\varepsilon)$ is the pair $(P_\mathcal{S}, Q_\mathcal{S})$ as defined below that provides a lower bound in the case of a single epoch.

$$P_\mathcal{S} := \sum_{s=1}^{S} \tfrac{1}{S} \cdot \mathcal{N}(2e_s, \sigma^2 I) \quad \text{and} \quad Q_\mathcal{S} := \sum_{s=1}^{S} \tfrac{1}{S} \cdot \mathcal{N}(e_s, \sigma^2 I).$$

ABLQ$_{\mathcal{S}^\circ}$ is an $E$-fold composition of the single-epoch mechanism. Hence by composition of dominating pairs, it follows that $\delta_{\mathcal{S}^\circ}(\varepsilon) \geq \max\{D_{e^\varepsilon}(P_{\mathcal{S}^\circ}\|Q_{\mathcal{S}^\circ}), D_{e^\varepsilon}(Q_{\mathcal{S}^\circ}\|P_{\mathcal{S}^\circ})\}$, where $P_{\mathcal{S}^\circ} := P_\mathcal{S}^{\otimes E}$ and $Q_{\mathcal{S}^\circ} := Q_\mathcal{S}^{\otimes E}$. However, it is tricky to directly identify an event $\Gamma$ for which the lower bound $P_{\mathcal{S}^\circ}(\Gamma) - e^\varepsilon Q_{\mathcal{S}^\circ}(\Gamma)$ is non-trivial and $P_{\mathcal{S}^\circ}(\Gamma)$, $Q_{\mathcal{S}^\circ}(\Gamma)$ are easy to compute. So in order to lower bound $D_{e^\varepsilon}(P_{\mathcal{S}^\circ}\|Q_{\mathcal{S}^\circ})$, below we construct a pair $(\tilde{P}_\mathcal{S}, \tilde{Q}_\mathcal{S})$ of discrete distributions such that $(P_\mathcal{S}, Q_\mathcal{S}) \succcurlyeq (\tilde{P}_\mathcal{S}, \tilde{Q}_\mathcal{S})$ and thus $(P_{\mathcal{S}^\circ}, Q_{\mathcal{S}^\circ}) \succcurlyeq (\tilde{P}_\mathcal{S}^{\otimes E}, \tilde{Q}_\mathcal{S}^{\otimes E})$.

For probability measures $P$ and $Q$ over a measurable space $\Omega$, and a finite partition[3] $\mathcal{G} = (G_1, \ldots, G_m)$ of $\Omega$, we can consider the discrete distributions $P^\mathcal{G}$ and $Q^\mathcal{G}$ defined over $\{1, \ldots, m\}$ such that $P^\mathcal{G}(i) = P(G_i)$ and $Q^\mathcal{G}(i) = Q(G_i)$. The post-processing property of DP implies:

**Proposition 3.6** (DP Post-processing [Dwork and Roth, 2014])**.** *For all partitions $\mathcal{E}$ of $\Omega$, it holds that $(P, Q) \succcurlyeq (P^\mathcal{E}, Q^\mathcal{E})$.*

We construct the pair $(\tilde{P}_\mathcal{S}, \tilde{Q}_\mathcal{S})$ by instantiating Proposition 3.6 with the set $\mathcal{G} = \{G_0, G_1, \ldots, G_{m+1}\}$ parameterized by a sequence $C_1 < \cdots < C_m$ of values defined as follows: $G_0 := \{w \in \mathbb{R}^S : \max_s w_s \leq C_1\}$, $G_i := \{w \in \mathbb{R}^S : C_i < \max_s w_s \leq C_{i+1}\}$ for $1 \leq i \leq m$ and $E_{m+1} := \{w \in \mathbb{R}^S : C_m < \max_s w_s\}$; in other words, $G_0 = \mathbb{R}^S \smallsetminus \Gamma_{C_1}$, $G_i = \Gamma_{C_i} \smallsetminus \Gamma_{C_{i+1}}$ for $1 \leq i \leq m$ and $G_{m+1} = \Gamma_{C_m}$.

**Theorem 3.7.** *For all $\sigma > 0$, $\varepsilon \geq 0$, all valid $n$, $b$, $T$, and any finite sequence $C_1, \ldots, C_m$ of values used to define $\tilde{P}_\mathcal{S}$, $\tilde{Q}_\mathcal{S}$ as above, it holds that*

$$\delta_{\mathcal{S}^\circ}(\varepsilon) \geq \max\{D_{e^\varepsilon}(\tilde{P}_\mathcal{S}^{\otimes E}\|\tilde{Q}_\mathcal{S}^{\otimes E}), D_{e^\varepsilon}(\tilde{Q}_\mathcal{S}^{\otimes E}\|\tilde{P}_\mathcal{S}^{\otimes E})\}.$$

---

[3]$G_i$'s are pairwise disjoint and $\cup_{i=1}^{m} G_i = \Omega$.

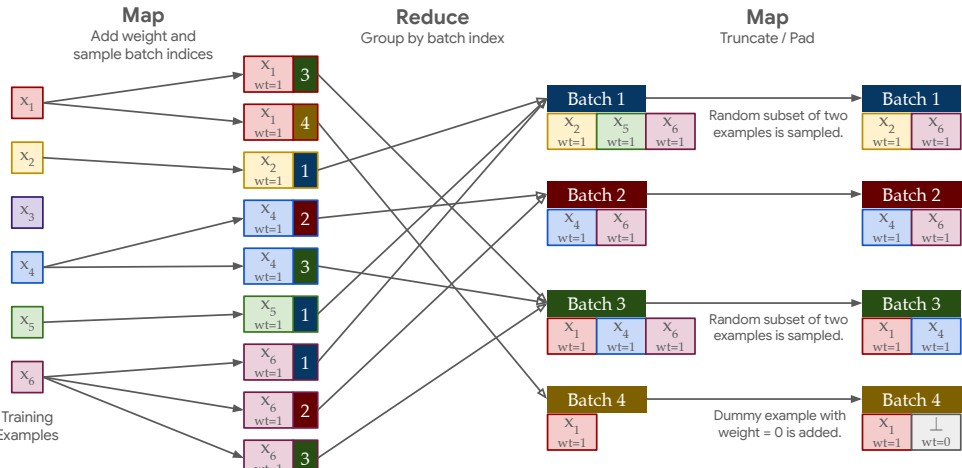

Figure 2: Visualization of the massively parallel computation approach for Poisson subsampling at scale. Consider 6 records $x_1, \ldots, x_6$ sub-sampled into 4 batches with a maximum batch size of $B = 2$. The Map operation adds a "weight" parameter of 1 to all examples, and samples indices of batches to which each example will belong. The Reduce operation groups by the batch indices. The final Map operation truncates batches with more than $B$ examples (e.g., batches 1 and 3 above), and pads dummy examples with weight 0 in batches with fewer than $B$ examples (e.g., batch 4 above).

We use the dp_accounting library [Google's DP Library., 2020] to numerically compute a lower bound on the quantity above, using PLD. In particular, we choose $C_1$ and $C_m$ such that $P_{\mathcal{S}}(G_0)$ and $P_{\mathcal{S}}(G_{m+1})$ are sufficiently small and choose other $C_i$'s to get a sufficiently fine discretization of the interval between $C_1$ and $C_m$.[4]

An illustration of these accounting methods is presented in Figure 3, which demonstrates the significant gap where the optimal $\sigma$ for dynamic/persistent shuffling is significantly larger than compared to Poisson subsampling, even when using an optimistic estimate for shuffling as above. We provide the implementation of our privacy accounting methods described above in an iPython notebook[5] hosted on Google Colab, executable using the freely available Python CPU runtime.

## 4 Experiments

We compare DP-SGD using the following batch sampling algorithms at corresponding noise scales:
- Deterministic batches (using nearly exact value of $\sigma_{\mathcal{D}}(\varepsilon, \delta)$ via Theorem 3.1),
- Truncated Poisson subsampled batches (using upper bound on $\sigma_{\mathcal{P}}(\varepsilon, \delta)$ via Theorem 3.3),
- Persistent shuffled batches (using lower bound on $\sigma_{\mathcal{S}\diamond}(\varepsilon, \delta)$ via Theorem 3.5), and
- Dynamic shuffled batches (using lower bound on $\sigma_{\mathcal{S}\circ}(\varepsilon, \delta)$ via Theorem 3.7).

As a comparison, we also evaluate DP-SGD with dynamic shuffled batches, but using noise that is an upper bound on $\sigma_{\mathcal{P}}(\varepsilon, \delta)$ (with no truncation, i.e. $B = \infty$), to capture the incorrect, but commonly employed approach in practice. Finally, in order to understand the impact of using different batch sampling to model training in isolation, we compare models trained with SGD under truncated Poisson subsampling, and dynamic and persistent shuffling without any clipping or noise. We use massively parallel computation (Map-Reduce operations [Dean and Ghemawat, 2004]) to generate batches with truncated Poisson subsampling in a scalable manner as visualized in Figure 2; the details, with a beam pipeline implementation, is provided in Appendix A.

---

[4]In our evaluation, we choose $C_1$ and $C_m$ to ensure $P_{\mathcal{S}}(G_0) + P_{\mathcal{S}}(G_{m+1}) \leq e^{-40}$. We chose other $C_i$'s to be equally spaced in between $C_1$ and $C_m$ with a gap of $\Delta \cdot \sigma^2$, where $\Delta$ is the desired discretization of the PLD. This heuristic choice is guided by the intuition that the privacy loss is approximately linear in $\max_t x_t/\sigma^2$, and thus the chosen gap means that this approximate privacy loss varies by $\Delta$ between buckets.

[5]https://colab.research.google.com/drive/1vCijMEQqRCm0x3EOUUKomcZnwx76sz64?usp=sharing

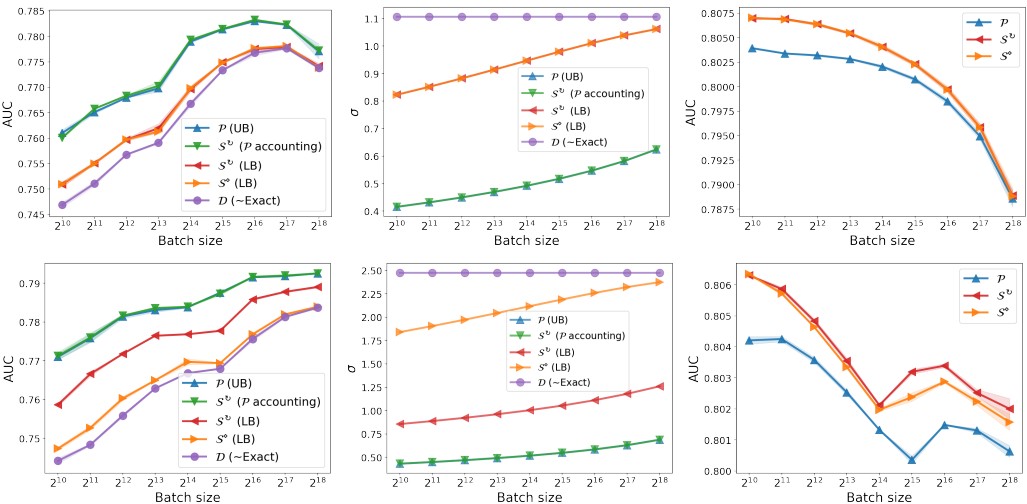

Figure 3: AUC (left) and bounds on $\sigma_{\mathcal{B}}$ values (middle) for $\varepsilon = 5, \delta = 2.7 \cdot 10^{-8}$ and using 1 epoch (top) and 5 epochs (bottom) of training on a linear-log scale; AUC (right) is with non-private training.

We run our experiments on the Criteo Display Ads pCTR Dataset [Jean-Baptiste Tien, 2014], which contains around 46 million examples from a week of Criteo ads traffic. Each example has 13 integer features and 26 categorical features, and the objective is to predict the probability of clicking on an ad given these features. We use the labeled training set from the dataset, split chronologically into a 80%/10%/10% partition of train/validation/test sets. We use the binary cross entropy loss and report the AUC on the labeled test split, averaged over three runs with different random seeds. These are plotted with error bars indicating a single standard deviation. We include more details about the model architectures and training in Appendix B.

We run experiments varying the (expected) batch size from $1\,024$ to $262\,144$, for both private training with $(\varepsilon = 5, \delta = 2.7 \cdot 10^{-8})$ and non-private training and with 1 or 5 epochs. We plot the results in Figure 3. As mostly expected, we observe that the model utility generally improves with smaller $\sigma$. Truncated Poisson subsampling performs similarly to dynamic shuffling for the same value of $\sigma$, although it performs worse for non-private training. The latter could be attributed to the fact that when using truncated Poisson subsampling, a substantial fraction[6] of examples are never seen in the training with high probability. However, this does not appear to significantly affect the private training model utility for the range of parameters that we consider, since we observe that truncated Poisson subsampling behaves similarly to dynamic shuffling with noise scale of $\sigma_{\mathcal{P}}(\varepsilon, \delta)$ (the values of $\sigma$ for "$\mathcal{P}$" and "$S^{\circlearrowright}$ ($\mathcal{P}$ accounting)" visually overlap in Figure 3; the values for $\mathcal{P}$ are only negligibly larger, since it accounts for truncation). Truncated Poisson subsampling performs better when compared to shuffling when the latter using our lower bound on $\sigma_{S^{\circlearrowright}}(\varepsilon, \delta)$, which suggests that shuffling with correct accounting (that is, with potentially even larger $\sigma$) would only perform worse.

We also run experiments with a fixed batch size $65\,536$ and varying $\varepsilon$ from 1 to 256, fixing $\delta = 2.7 \cdot 10^{-8}$. We plot the results and the corresponding $\sigma$ values in Figure 4. We again observe that shuffling (with our lower bound accounting) performs worse than truncated Poisson subsampling in the high privacy (low $\varepsilon$) regime, but performs slightly better in the low privacy (high $\varepsilon$) regime (this is because at large $\varepsilon$ the noise required under Poisson subsampling is in fact larger than that under shuffling). Moreover, we observe that shuffling performs similarly to truncated Poisson subsampling when we use similar value of $\sigma$, consistent with our observations from Figure 3.

Finally, as a comparison, we compute the upper bounds on $\sigma_{\mathcal{S}}(\varepsilon, \delta)$ via the privacy amplification by shuffling bounds by Feldman et al. [2021]. We find that these bounds tend to be vacuous in many regime of parameters, namely they are no better than $\sigma_{\mathcal{D}}(\varepsilon, \delta)$, which is clearly an upper bound on $\sigma_{\mathcal{S}}(\varepsilon, \delta)$. Details are provided in Appendix C.

---

[6]The expected number of examples that are never used even once during training is at least $(1 - \frac{b}{n})^T$, which approaches $e^{-E}$ in the limit as $b/n \to 0$, for a fixed number of epochs $E = bT/n$.

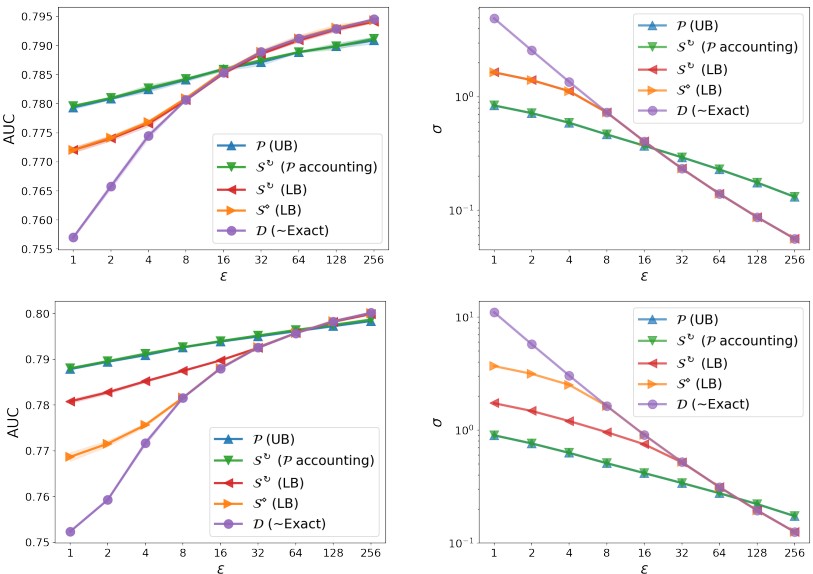

Figure 4: AUC (left) and $\sigma$ values (right) with varying $\varepsilon$, fixing $\delta = 2.7 \cdot 10^{-8}$ and using (top) 1 epoch and (bottom) 5 epochs of training. $\sigma$ is in log scale to highlight the differences at high $\varepsilon$.

## 5 Conclusion

We provide new lower bounds on the privacy analysis of Adaptive Batch Linear Query mechanisms, under persistent and dynamic shuffling batch samplers, extending the prior work of Chua et al. [2024] that analyzed the single epoch case. Our lower bound method continues to identify separations in the multi-epoch setting, showing that the amplification guarantees due to even dynamic shuffling can be significantly limited compared to the amplification due to Poisson subsampling in regimes of practical interest.

We also provide evaluation of DP-SGD with various batch samplers with the corresponding privacy accounting, and propose an approach for implementing Poisson subsampling at scale using massively parallel computation. Our findings suggest that with provable privacy guarantees on model training, Poisson-subsampling-based DP-SGD has better privacy-utility trade-off than shuffling-based DP-SGD in many practical parameter regimes of interest, and in fact, essentially match the utility of shuffling-based DP-SGD at the same noise level. Thus, we consider Poisson-subsampling-based DP-SGD as a viable approach for implementing DP-SGD at scale, given the lower bound on the privacy analysis when using shuffling.

Several interesting directions remain to be investigated. Firstly, our technique only provides a *lower bound* on the privacy guarantee when using persistent / dynamic shuffled batches. While some privacy amplification results are known [Feldman et al., 2021, 2023], providing a tight (non-vacuous) upper bound on the privacy guarantee in these settings remains an open challenge. This can be important in regimes where shuffling does provide better privacy guarantees than Poisson subsampling.

Another important point to note is that persistent and dynamic shuffling are not the only forms of shuffling used in practice. For example, methods such as `tf.data.Dataset.shuffle` or `torchdata.datapipes.iter.Shuffler` provide a uniformly random shuffle, only when the size of its "buffer" is larger than the dataset. Otherwise, for buffer size $b$, it returns a random record among the first $b$ records, and immediately replaces it with the next record $((b + 1)$th in this case), and repeats this process, which leads to an asymmetric form of shuffling. Such batch samplers merit more careful privacy analysis.

#### Acknowledgements

We would like to thank Charlie Harrison and Ethan Leeman for valuable discussions, as well as anonymous reviewers for their thoughtful feedback that helped improve the quality of the paper.

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

# A  Massively Parallel Implementation of Truncated Poisson Subsampling

We use massively parallel computation to generate batches with truncated Poisson subsampling in a scalable manner. Given the input parameters $b$, $B$, $T$, and $n$, we first compute the maximum batch size $B$ such that $\Psi(n, b, B) \cdot T \cdot (1 + e^\varepsilon) \leq 10^{-5} \cdot \delta$. For each example in the input dataset, we generate a list of batches that the example would be in when sampled using Poisson subsampling. While a naive implementation would sample $T$ Bernoulli random variables with parameter $b/n$, this can be made efficient by sampling the indices of the batches containing the examples directly, since the difference between two consecutive such indices is distributed as a geometric random variable with parameter $b/n$. We then group the examples by the batches, and subsample each batch uniformly, without replacement, to obtain a batch of size at most $B$. For batches with size smaller than $B$, we pad the batch with examples such that every batch has size $B$. In order to differentiate the padding examples from the non-padding examples, we add a weight to all the examples, where the non-padding examples have weight $1$ and the padding examples have weight $0$. During the training, we use a weighted loss function using these weights, such that the padding examples do not have any effect on the training loss.

We include a code snippet for implementing truncated Poisson subsampling using massively parallel computation. This is written using Apache beam [Apache Beam] in Python, which can be implemented on distributed platforms such as Apache Flink, Apache Spark, Google Cloud Dataflow.

```python
import apache_beam as beam
import numpy as np
import tensorflow as tf

class PoissonSubsample(beam.PTransform):
  """Generate batches of examples using poisson subsampling.

  Attributes:
    max_batch_size: Maximum batch size.
    num_batches: Number of batches.
    subsampling_probability: Probability of sampling each example in each batch.
    sample_size: Number of samples to sample at a time.
  """

  def __init__(
      self,
      max_batch_size: int,
      num_batches: int,
      subsampling_probability: float,
      sample_size: int,
  ):
    self._max_batch_size = max_batch_size
    self._num_batches = num_batches
    self._subsampling_probability = subsampling_probability
    self._sample_size = sample_size
    self._rng = np.random.default_rng()

  def get_batch_indices(self):
    """Returns the indices of the batches that an example is in.

    Assuming that an example is sampled in each batch using poisson subsampling,
    return the list of indices of the batches that the example is in.
    """
    largest_batch_index = 0
    batch_indices = np.array([])
    if self._subsampling_probability == 0.0:
      return batch_indices
    while largest_batch_index < self._num_batches:
      # Sample batches using geometric distribution
      geometric_samples = self._rng.geometric(
          p=self._subsampling_probability, size=self._sample_size
```

```python
    )
    batch_indices = np.concatenate(
        (batch_indices, largest_batch_index + np.cumsum(geometric_samples))
    )
    largest_batch_index = batch_indices[-1]
  return batch_indices[batch_indices <= self._num_batches]

def _add_padding(self, batch):
  """Returns batch padded to max_batch_size with padding examples.

  Pads input batch to size max_batch_size by adding padding examples. The
  padding examples are specified by adding a weight to all the examples, where
  padding examples have weight 0 and non-padding examples have weight 1.

  Args:
    batch: A tuple of (batch_id, list of examples)
  """
  batch_id, examples = batch
  examples = list(examples)
  if len(examples) < self._max_batch_size:
    padding_example = tf.train.Example()
    padding_example.CopyFrom(examples[0])
    padding_example.features.feature['weight'].float_list.value[:] = [0.0]
    examples.extend(
        [padding_example] * (self._max_batch_size - len(examples))
    )
  return (batch_id, examples)

def expand(self, pcoll):
  def generate_batch_ids(example):
    # Convert to pairs consisting of (batch id, example)
    for batch_id in self.get_batch_indices():
      yield (int(batch_id), example)

  def _add_weights(example):
    # Add weight with value 1 to indicate non-padding examples
    weighted_example = tf.train.Example()
    weighted_example.CopyFrom(example)
    weighted_example.features.feature['weight'].float_list.value[:] = [1.0]
    return weighted_example

  # Group elements into batches keyed by the batch id
  grouped_pcoll = (
      pcoll
      | 'Add weights' >> beam.Map(_add_weights)
      | 'Key by batch id' >> beam.FlatMap(generate_batch_ids)
      | 'Sample up to max_batch_size elements per batch'
      >> beam.combiners.Sample.FixedSizePerKey(self._max_batch_size)
      | 'Add padding' >> beam.Map(self._add_padding)
  )
  return grouped_pcoll
```

# B Training details

We use a neural network with five layers and $\sim$78M parameters as the model. The first layer consists of feature transforms for each of the categorical and integer features. Categorical features are mapped into dense feature vectors using an embedding layer, where the embedding dimensions are fixed at $48$. We apply a log transform for the remaining integer features, and concatenate all the features together. The next three layers are fully connected layers with $598$ hidden units each and a ReLU activation function. The last layer consists of a fully connected layer which gives a scalar logit prediction.

We use the Adam or Adagrad optimizer with a base learning rate in $\{0.0001, 0.0005, 0.001, 0.005, 0.01, 0.05, 0.1, 0.5, 1\}$, which is scaled with a cosine decay, and we tune the norm bound

$C \in \{1, 5, 10, 50\}$. For the experiments with varying batch sizes, we use batch sizes that are powers of 2 between $1\,024$ and $262\,144$, with corresponding maximum batch sizes $B$ in $\{1\,328, 2\,469, 4\,681, 9\,007, 17\,520, 34\,355, 67\,754, 134\,172, 266\,475\}$. For the experiments with varying $\varepsilon$, we vary $\varepsilon$ as powers of 2 between 1 and 256, with batch size $65\,536$ and corresponding maximum batch sizes $B$ in $\{67\,642, 67\,667, 67\,725, 67\,841, 68\,059, 68\,449, 69\,106, 70\,156, 71\,760\}$. With these choices of the maximum batch sizes, the $\sigma$ values for truncated Poisson subsampling are only slightly larger than without truncation, as we observe from the nearly overlapping curves in Figure 3 and Figure 4. The training is done using NVIDIA Tesla P100 GPUs, where each epoch of training takes 1-2 hours on a single GPU.

## C  Privacy Amplification by Shuffling

We evaluate the upper bounds on $\sigma_{\mathcal{S}}(\varepsilon, \delta)$ via privacy amplification by shuffling results of Feldman et al. [2021], as applied in the context of our experiments in Figure 3. In particular, their Proposition 5.3 states that if the unamplified (Gaussian) mechanism satisfies $(\varepsilon_0, \delta_0)$-DP, then $\mathsf{ABLQ}_{\mathcal{S}}$ with $T$ steps (in a single epoch setting) will satisfy $(\varepsilon, \eta + O(e^{\varepsilon}\delta_0 n))$-DP for any $\eta \in [0, 1]$ and

$$\varepsilon = O\left( (1 - e^{-\varepsilon_0}) \left( \frac{\sqrt{e^{\varepsilon_0} \log(1/\eta)}}{\sqrt{T}} + \frac{e^{\varepsilon_0}}{T} \right) \right).$$

They also obtain a tighter numerical bound on $\varepsilon$ with an implementation provided in a GitHub repository.[7]

We evaluate their bounds in an optimistic manner. Namely, for a given value of $\delta$, we compute an optimistic estimate on $\varepsilon$ compared to the bound above by setting $\delta_0 = \delta/n$ and setting $\varepsilon_0 = \varepsilon_{\mathcal{D}}(\delta_0)$, and set $\eta = \delta$ (note that this is optimistic because the above proposition requires setting $\delta = \eta + O(e^{\varepsilon}\delta_0 n)$, which is larger than the $\delta$ we are claiming). We use the numerical analysis method provided in the library by Feldman et al. [2021] to compute $\sigma_{\mathcal{S}}$ (via a binary search on top of their method to compute an upper bound on $\varepsilon_{\mathcal{S}}$), and plot it in Figure 5, along with the lower bound on $\sigma_{\mathcal{S}}$ as obtained by Chua et al. [2024], as well as $\sigma_{\mathcal{D}}$. We find that the bounds by Feldman et al. [2021] are vacuous for batch sizes $2\,048$ and above, in that they are even larger than the bounds without any amplification.

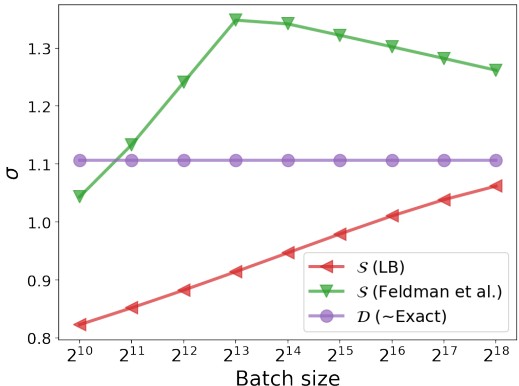

Figure 5: Comparison of an optimistic estimate of the upper bound on $\sigma_{\mathcal{S}}$ from Feldman et al. [2021] against the lower bound on $\sigma_{\mathcal{S}}$ in Theorem 3.5 and $\sigma_{\mathcal{D}}$.

## D  $\sigma$ with varying epochs

We include a comparison of the $\sigma$ values for varying numbers of epochs, to show how the same trends hold beyond the 1 and 5 epoch regimes.

---

[7] https://github.com/apple/ml-shuffling-amplification

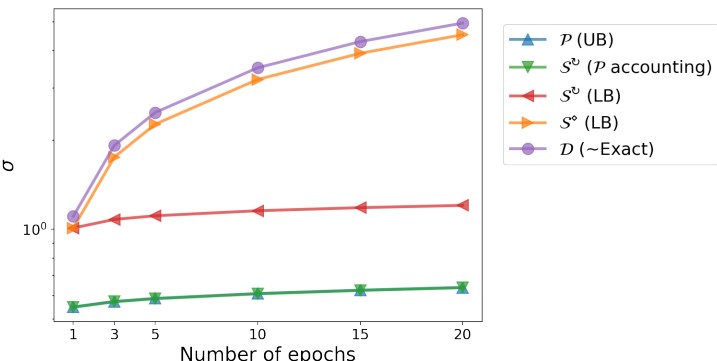

Figure 6: $\sigma_{\mathcal{B}}$ values with varying numbers of epochs, fixing $\varepsilon = 5$, $\delta = 2.7 \cdot 10^{-8}$, and batch size 65 536.

