# OpenReview forum: "Scalable DP-SGD: Shuffling vs. Poisson Subsampling"
_NeurIPS.cc/2024/Conference — NeurIPS 2024 poster_

### Official Review · Reviewer_k7t2 · 2024-06-23

**Soundness:** 4
**Presentation:** 4
**Contribution:** 2
**Rating:** 5
**Confidence:** 4

**Summary:**

DP-SGD is one of the must common algorithms currently deployed for performing machine learning tasks while maintaining differential privacy. As the "S" in its name reminds us, the gradient is not computed over the full dataset at each turn, but instead over a subset sampled from it. The commonly used privacy analysis typically relies on Poisson sub-sampling, whereby each element is independently added to each batch with some constant probability. In contrast, due to technical challenges stemming from the fact the Poisson sub-sampled batch size is not constant, this is typically implemented by first shuffling the full dataset, then batching it into subsequent subsets.

Unfortunately, there is a gap between existing upper bounds on the privacy loss of the shuffle version and those of the sub-sampling. Recently Lynn et al. [1] provided a lower bound on the privacy of the shuffled version, which shows a clear gap between the two settings in some parameter regimes (mainly low local privacy).

This paper continues and extends [1] work in two ways:
1. It extends the analysis to multi-epoch setting
2. It proposes a variant of Poisson sub-sampling with effective constant batch size, by paying a small price in the privacy failure probability $\delta$
The authors also perform several empirical experiments to compare the privacy and accuracy of various versions of private training.

[1] Lynn Chua, Badih Ghazi, Pritish Kamath, Ravi Kumar, Pasin Manurangsi, Amer Sinha, and Chiyuan Zhang. How Private is DP-SGD? In International Conference on Machine Learning, ICML (to appear), 2024.

**Strengths:**

This paper is clearly written, and adds to our understanding of the toolkit one can use when attempting to learn privately. The proposed truncated sub-sampling might prove a useful alternative to the existing commonly used shuffle method.

**Weaknesses:**

While this work is clearly written and provides useful empirical and theoretical tools, its novelty is somewhat limited.
As mentioned, this is a follow-up work to the one by Lynn et al. The extension of their analysis to multi epochs is relatively straight forward, and so is the analysis of the truncated Poisson sub-sampling.

**Questions:**

Can the authors add a comparison of the the various privacy analyses to the upper bounds provided by existing literature on general shuffle model, such as the one provided by Feldman et al. [2]?

Additionally, to the best of my understanding, amplification by sub-sampling a constant size batches without replacement (per sampling)  is will understood, and enjoys privacy guarantees that are comparable to those of Poisson sub-sampling, up to a factor 2, depending on the exact neighboring notion. What is the advantage of the proposed method relative to the WOR sampling option?

[2] Feldman, V., McMillan, A., and Talwar, K. Hiding among the clones: A simple and nearly optimal analysis of privacy amplification by shuffling. In FOCS, pp. 954–964, 2021

---

> ### Author Rebuttal · Authors · 2024-08-06
>
> We thank the reviewer for their thoughtful comments.
>
> > add a comparison of the the various privacy analyses to the upper bounds provided by existing literature on general shuffle model
>
> While the amplification bounds for shuffling such as by Feldman et al. provide upper bounds, we did not consider it because
> * our focus in this work was primarily to compare _lower bounds_ for Shuffle DP-SGD with _upper bounds_ for Poisson DP-SGD, and
> * in the regime of small noise multipliers, where we see that our lower bound on the noise multiplier of Shuffle DP-SGD is quite close to that of Deterministic (no amplification), these upper bounds on amplification by shuffling have to necessarily be near-vacuous (cannot be much better than deterministic).
>
> Nevertheless, this is a valuable point, and we will include at least the noise multipliers obtained via these amplification upper bounds for comparison in a future revision.
>
> > sampling without replacement (Similar point raised by Reviewer 1Uvo)
>
> Indeed, sampling without replacement (i.e., sampling fixed-sizes batches uniformly and independently at random) can also be implemented using the massively parallel computation approach we propose. We did not however consider it in our evaluation however since the privacy guarantees of that are worse than that of Poisson subsampling. Similar to your point, as noted in this recent work of [Lebeda et al.](https://arxiv.org/abs/2405.20769), the noise scale required for this method is twice that required for Poisson subsampling. We will revise the paper to include a discussion of this alternative approach.

---

> > ### Comment · Reviewer_k7t2 · 2024-08-09
> >
> > I thank the authors for their response.
> >
> > As previously mentioned, the comparison of lower bounds for the shuffle model to upper bounds for the Poisson sub-sampling model were already performed by Chia et al. One of the main contributions of this current work as stated by the authors, is the proposal of the truncated Poisson sub-sampling model, to remedy the existing gap in private SGD based training. I find it hard to asses the importance of this contribution, where the upper bounds for two existing alternatives (shuffle and constant size sub-sampling) where not considered, and might very well provide accuracy results that are comparable to the new proposed technique.
> >
> > Additionally, I would recommend the authors better clarify in the notations which lines correspond to upper bounds and which ones to lower bounds. This seemed to cause confusion for some of the reviewers, and might affect the readers as well. In fact, the comparison discussed in line 284 creates the impression that the two compared lines represent the same quantity while - to the best of my understanding - they do not.

---

> > > ### Author Response · Authors · 2024-08-13
> > >
> > > > better clarify in the notations which lines correspond to upper bounds and which ones to lower bounds
> > >
> > > Just to clarify, the noise multiplier for $\cal D$ is essentially exact, the noise multipliers for $\cal P$ (as well as ${\cal S}^{\circlearrowright}$ with $\cal P$ accounting) correspond to upper bounds, and the noise multipliers for ${\cal S}^{\circlearrowright}$ and ${\cal S}^\diamond$ are lower bounds. Thanks for the feedback, we will incorporate this in the revision.

---

### Official Review · Reviewer_9tRV · 2024-07-08

**Soundness:** 4
**Presentation:** 3
**Contribution:** 4
**Rating:** 7
**Confidence:** 4

**Summary:**

This paper provides theoretical and empirical analyses of three different DP-SGD minibatch sampling schemes, and also implement an efficient beam pipeline for using Poisson sampling in practice via a truncation-based approach.  Theoretically, they derive new lower bounds for the privacy accounting / noise calibration of shuffling-based DP-SGD, and truncated Poisson-sampling-based DP-SGD.  Empirically, they show (1) for the same noise multiplier, shuffling does better (in terms of AUC) than Poisson sampling and (2) for the same privacy budget, Poisson sampling does better, mainly due to the more favorable privacy accounting.  In one experiment (Fig 3 upper left) they show that shuffling can sometimes be better than Poisson sampling for small epsilon.

**Strengths:**

* Addresses an important gap in many works on DP-SGD in a principled manner.
* Clearly demonstrates the benefits of Poisson sampling over shuffling with correct / tight privacy accounting.
* Shows how to implement Poisson sampling within the typical constraints of an ML pipeline via truncation + padding + offline beam job to order the examples.  This is crucial to get the best privacy / utility trade-offs and formal privacy guarantees in production DP applications.  The shuffling ~=~ sampling approximation may still make sense in research applications.
* Experiments compare the methods you'd expect to see (normalized by sigma and normalized by privacy budget).

**Weaknesses:**

There are a few missing pieces that I think would strengthen the paper and hopefully not require too much extra work:

1) Thm 3.3 uses the bound from Prop 3.2, and may not be the tightest possible result.  Would be good to quantify the gap between Poisson sampling and Truncated Poisson sampling in terms of the noise multiplier you obtain for each, fixing p for Poisson sampling and B for the Truncated Poisson sampling so that the expected batch sizes of Poisson sampling matches the physical batch sizes of truncated Poisson.
2) There is not enough discussion or analysis on how to configure the Truncated Poisson parameters.  For a fixed physical batch size, privacy budget, and other relevant parameters (number of iterations), can you give a simple procedure to find the best sampling probability to use in terms of the expected signal to noise ratio?
3) Do the same findings hold up beyond the 1- and 5-epoch regimes? Would be nice to include a comparison where Epochs is varied on the x-axis and sigma is varied on the y-axis (no need to do new experiments on real data which might be very expensive).

Minor:

4) In plots you should evenly space out the independent variables.  If plotting in log space, evenly space them in log space (e.g., Batch Size = 1024, 2048, 4096, ..., 262144 and epsilon = 1, 2, 4, 8, ..., 16).  Also the legend is a bit small, and shared across plots.  Consider making 1-row 5-column legend spanning or otherwise increasing the font size.

**Questions:**

1) In Fig 2, it seems like P corresponds to  truncated Poisson sampling, but green line corresponds to non-truncated Poisson accounting.  Why do these lines line up in the middle plots then?

---

> ### Author Rebuttal · Authors · 2024-08-06
>
> We thank the reviewer for their thoughtful comments.
>
> > quantify the gap between Poisson sampling and Truncated Poisson sampling
>
> While it is likely that our analysis for truncated Poisson sampling is not optimal, it is reasonable enough in practice. The loss due to truncation is very minimal to the privacy accounting as we allocate only $0.1 \delta$ for the truncation part, as seen by how the noise multiplier for Poisson subsampling with and without truncation are quite similar (seen in Fig. 2, 3). Note that the trade-off here is only between privacy and computation: larger maximum batch size means accounting is closer to (untruncated) Poisson subsampling, at the cost of increased computation during model training.
> Thus, the optimal accounting for truncated Poisson subsampling would at best allow us to use a slightly smaller maximum batch size, which would only slightly reduce the computational cost of training. Since this was not central to our work, we did not optimize the analysis to the best possible. Thanks for raising this, and we will consider adding this to the discussion.
>
> > find the best sampling probability to use in terms of the expected signal to noise ratio
>
> While we do not optimize this theoretically, this was indeed our motivation for considering different batch sizes in Figure 2.
>
> >  include a comparison where Epochs is varied on the x-axis and sigma is varied on the y-axis
>
> We provide $\sigma$ values for different accounting methods in the table below, as well as in a plot in Figure 1 of the attached pdf. Note that the values for the $\cal P$ sampler are upper bounds, whereas the values for $\cal S$ samplers are lower bounds. Again observe that the values for $\cal P$ (truncated Poisson subsampling) are very close to ${\cal S}^{\circlearrowright}$ ($\cal P$ accounting), the latter corresponding to untruncated Poisson subsampling.
>
> | Epochs | $\cal P$ | ${\cal S}^\circlearrowright$ ($\cal P$ accounting) |  ${\cal S}^\circlearrowright$ | ${\cal S}^{\diamond}$ | $\cal D$ |
> |---|---|---|---|---|---|
> | 1 | 0.608 | 0.606 | 1.053 | 1.053 | 1.106 |
> | 3 | 0.643 | 0.642 | 1.181 | 1.825 | 1.916 |
> | 5 | 0.665 | 0.663 | 1.227 | 2.356 | 2.474 |
> | 10 | 0.702 | 0.701 | 1.292 | 3.331 | 3.499 |
> | 15 | 0.730 | 0.729 | 1.334 | 4.080 | 4.285 |
> | 20 | 0.755 | 0.753 | 1.366 | 4.711 | 4.948 |
>
> Observe that the values of Poisson subsampling with truncation are slightly larger than that without truncation.
>
> > evenly space out the independent variables
>
> Thanks for the suggestion. We will modify the plots accordingly.
>
> > Why do these lines line up in the middle plots
>
> As argued above, the additional privacy loss due to truncation of the batch in Poisson subsampling is minimal even with our sub-optimal analysis, and so while the values with and without truncation are not exactly the same, the difference is negligible compared to the scale of the rest of the plot.

---

> > ### Comment · Reviewer_9tRV · 2024-08-12
> >
> > Apologies I am just looking at this carefully now.  For the table you shared, those noise multipliers appear to not account for the differing expected batch sizes, is that correct?
> >
> > For P, I would like to see noise multiplier / (expected batch size)
> >
> > For S (with P accounting) I would like to see noise multipliers / (physical batch size)
> >
> > Please update and send back ASAP, I would like to follow-up after seeing those numbers.

---

> ### Author Response · Authors · 2024-08-12
> **Response to Official Comment by Reviewer 9tRV**
>
> We would like to clarify that for the table we shared, the expected batch sizes for Poisson subsampling are the same as the physical batch size for Shuffle and Deterministic; by "${\cal S}^{\circlearrowright}$ (with $\cal P$ accounting)", we mean the same method as ${\cal S}^{\circlearrowright}$, but using the noise multiplier assuming Poisson subsampling (without any truncation). While truncation in Poisson subsampling does reduce the expected batch size by a small amount, this is quite negligible and hence the normalization would be by approximately the same quantity. In particular, we use the values of $\varepsilon = 5$, $\delta = 2.7 \cdot 10^{-8}$ and (expected) batch size $b = 200000$. Please let us know if there are still any further questions.

---

> > ### Comment · Reviewer_9tRV · 2024-08-12
> >
> > I see, I would have thought the plot would be normalized for physical batch size.
> >
> > For Truncated Poisson sampling what is the physical batch size you use then?

---

> ### Author Response · Authors · 2024-08-12
>
> For truncated Poisson subsampling, we use a maximum batch size as specified below, which includes the dummy examples added to ensure that all batches have the same size. The normalization is still by expected batch size (before truncation), which is $b = 200,000$.
>
> | Epochs | 1 | 3 | 5 | 10 | 15 | 20 |
> |---|---|---|---|---|---|---|
> | **Max Batch size** | 203288 | 203353 | 203383 | 203423 | 203446 | 203463 |

---

> > ### Comment · Reviewer_9tRV · 2024-08-12
> >
> > I see, so only a ~1.7% "cost" -- perhaps there is not much to be gained by optimizing the parameters.  Thanks for clarification, would be good to include a version of this table in the paper or appendix.

---

### Official Review · Reviewer_1Uvo · 2024-07-12

**Soundness:** 4
**Presentation:** 3
**Contribution:** 3
**Rating:** 6
**Confidence:** 3

**Summary:**

The paper focuses on practical implementations of DP training of ML models at scale in the multi-epoch setting. The contribution is two-fold: the paper gives a rigorous analysis for a practical version of Poisson subsampled DP-SGD where the batch size is upper bounded, and proposes lower $(\varepsilon,\delta)$-bounds for the shuffled Gaussian mechanism. Both of these contributions require a careful theoretical analysis which the paper succeeds in carrying out such that the bounds are sharp: there is no big slack in the approximative Poisson subsampling upper bound and the lower bounds is able to illustrate that the shuffling mechanism generally leads to a worse privacy-utility trade-off than the Poisson subsampled Gaussian mechanism. So the message becomes clear: although the shuffling + disjoint batch training is practical, the practical version the paper proposes will likely lead to a better privacy-utility trade-off. The paper builds upon the paper (Chua et al., ICML 2024) and borrows several results from it.

**Strengths:**

- Clear message and a well-written paper on a timely topic. DP training of ML models is becoming more popular and these practical aspects need to be addressed.

- Elegant theoretical analysis for the approximative Poisson mechanism and for the lower bounds of the shuffled Gaussian mechanism.

**Weaknesses:**

- The paper heavily builds upon the previous work by Chua et al. (ICML 2024), and the novelty is relatively thin, although the paper succeeds in delivering a clear message.

- The privacy amplification by iteration analysis - type of analysis is not mentioned. Although the current analyses in that line of work are only applicable to convex problems, they give $(\varepsilon,\delta)$-DP bounds for e.g. noisy cyclic GD (Bok et al., ICML 2024) that lead to models whose privacy-utility trade-off is competitive with the DP-SGD trained models (see e.g. experiments by Bok et al.).

**Questions:**

- You mention that the analysis for the shuffled Gaussian mechanism is an open problem. Why do you think this is an important open problem if the lower bounds for the shuffled Gaussian mechanism already indicate that DP-SGD give much better privacy-utility trade-off? In which scenarios would the upper bounds be useful?

- Why do you only consider the Poisson subsampling? Is there something particular in the subsampling without replacement that makes it less suitable to the large scale setting? In that case you would not need to have those approximations that you carry out to limit the batch size.

**Limitations:**

Yes.

---

> ### Author Rebuttal · Authors · 2024-08-06
>
> We thank the reviewer for their thoughtful comments.
>
> > novelty over prior work [Similar to Reviewer vHjL]
>
> While it may seem that our privacy analysis used some standard techniques (discretization, post-processing and PLD accounting), it was a priori unclear if such a simple method would be effective at providing strong lower bounds, let alone what such a method (e.g. the choice of discretization sets) would be. We would like to note that the prior work of [Chua et al](https://arxiv.org/abs/2403.17673) explicitly notes in Section 5, that _[their] approach is limited to a “single epoch” mechanism ... Extending [their] approach to multiple epochs will be interesting._ Moreover, while they are primarily focused on theoretical analysis, we also study practical implementations and evaluations on real world problems to verify the empirical feasibility of the proposed methods.
>
> > privacy amplification by iteration is not mentioned.
>
> Understanding when privacy amplification by iteration applies remains an interesting direction, but as of now there is no evidence that it can provide any improvements in the general (non-convex) setting. In fact, a recent [work](https://arxiv.org/abs/2407.06496) suggests that such an improvement might not be possible for general non-convex losses. We will include a discussion on the same in a revision.
>
> > Importance of upper bounds for the shuffled DP-SGD?
>
> This would be relevant in a regime where Shuffle-DP-SGD performs better than Poisson subsampling. In this case, we would need an “upper-bound” accounting method to report a correct DP guarantee. But we agree that it is less relevant in regimes where Shuffle DP-SGD has worse model utility.
>
> > subsampling without replacement (Similar point raised by Reviewer k7t2)
>
> Sampling without replacement (i.e., sampling fixed-sizes batches uniformly and independently at random) can also be implemented using a variant of the massively parallel computation approach we propose. We did not consider it in our evaluation however since the privacy guarantees of that are worse than that of Poisson subsampling. In particular, as noted in this recent work of [Lebeda et al.](https://arxiv.org/abs/2405.20769), the noise scale required for this method is twice that required for Poisson subsampling. We will revise the paper to include a discussion of this alternative approach.

---

> > ### Comment · Reviewer_1Uvo · 2024-08-08
> >
> > Thank you for the replies! I will keep my score. Please do add the discussion about subsampling without replacement. Please take into account also the conjecture by [Lebeda et al.](https://arxiv.org/pdf/2405.20769) that subsampling without replacement behaves similarly both for add/remove and substitute neighborhood relations.

---

### Official Review · Reviewer_vHjL · 2024-07-17

**Soundness:** 2
**Presentation:** 3
**Contribution:** 2
**Rating:** 3
**Confidence:** 4

**Summary:**

This paper investigates the utility of models trained with DP-SGD based on previous findings on the gap in privacy guarantee between shuffled batch sampling (commonly used in practice) and Poisson subsampling (used in theoretical analysis) for private training. A scalable implementation of Posson subsampling is proposed via truncation and parallelization, provided with lower bounds of the privacy guarantee via multi-epoch ABLQ with shuffling.

**Strengths:**

- This paper studies the implementation of subsampling with associated privacy accounting in private training and extends the results of ABLQ for multiple epochs.
- A scalable implementation of Possion subsampling is proposed, provided with lower bounds on its privacy guarantee.

**Weaknesses:**

- The techniques for extending ABLQ from multi-batch to multi-epoch are straightforward.
- It is difficult to distinguish from the existing results of Chua et al. [2024], given much of the material in Sec. 2 and Sec. 3 overlaps and is sometimes verbatim.
- Different types of batch sampling with different privacy accounting are evaluated. Still, all of them underestimate the privacy loss and cannot rigorously reflect the model utility under the claimed $(\epsilon, \delta)$.

**Questions:**

- The authors mentioned that the Opacus library supports Possion subsampling but might not be used for training on massive datasets. Have the authors tested the implementation from Opacus on the Criteo dataset? If so, how large is a batch it can support (from 1000 to 200,000)? Is this the major obstacle to be included as a reference for Fig.2 and 3?
- Is the target batch size $b$ still used to average the summed noisy gradients for the truncated Poisson Batch sampler? Could the authors elaborate on how the size of truncated batches would affect the tracking of privacy loss, especially in Theorem 3.3?
- Line 254 -255, could the authors provide more details on selecting $C_i$ in practice? And what is the computation complexity regarding this part?

**Limitations:**

See W2,3 and Q1

---

> ### Author Rebuttal · Authors · 2024-08-06
>
> We thank the reviewer for their thoughtful comments.
>
> > techniques for extending to multi-epoch are straightforward. [Similar to Reviewer 1Uvo]
>
> While it may seem that our privacy analysis used some standard techniques (discretization, post-processing and PLD accounting), it was a priori unclear if such a simple method would be effective at providing strong lower bounds, let alone what such a method (e.g. the choice of discretization sets) would be. We would like to note that the prior work of [Chua et al](https://arxiv.org/abs/2403.17673) explicitly notes in Section 5, that _[their] approach is limited to a “single epoch” mechanism ... Extending [their] approach to multiple epochs will be interesting._ Moreover, while they are primarily focused on theoretical analysis, we also study practical implementations and evaluations on real world problems to verify the empirical feasibility of the proposed methods.
>
> > material overlap in Sec. 2 and 3.
>
> Section 2 consists mainly of definitions (some standard and some from [Chua et al](https://arxiv.org/abs/2403.17673)), resulting in the overlap; please note that we have cited them in all relevant places. Nevertheless, thank you for bringing this to our attention and we will keep this in mind as we revise the paper to highlight our contributions better.
>
> > Different types of batch sampling with different privacy accounting are evaluated ... all of them underestimate the privacy loss and cannot rigorously reflect the model utility under the claimed $(\epsilon, \delta)$.
>
> We would like to clarify a possible misunderstanding here, that for Poisson subsampling based DP-SGD, we use the `dp_accounting` library that _overestimates_ the privacy loss, and for Shuffle based DP-SGD, our method _underestimates_ the privacy loss. So the way to interpret our results is that: whenever Poisson subsampling based DP-SGD out-performs Shuffle-based DP-SGD, this would hold even if we use the optimal accounting for each method.
>
> > Have the authors tested the implementation from Opacus on the Criteo dataset?
>
> Using the implementation from Opacus requires efficient random access to all the data points (e.g. loading the entire dataset in memory), which is not always feasible depending on the machine and configuration, given that the Criteo dataset takes 49GB of storage. We believe such a comparison does not provide too much scientific value, because for small datasets that can fit in memory, the Opacus solution and our method would perform similarly because they are mathematically equivalent. But in contrast to our methods, to the best of our knowledge, it is not straightforward how to scale Opacus to larger datasets.
>
> >Is the target batch size b still used to average the summed noisy gradients for the truncated Poisson Batch sampler? Could the authors elaborate on how the size of truncated batches would affect the tracking of privacy loss, especially in Theorem 3.3?
>
> We average the summed noisy gradients using the expected batch size; in any case, this is simply a scaling factor that can be assimilated in the learning rate, hence it is not important whether we normalize by the expected or maximum batch size. There is a privacy-vs-computation tradeoff in the choice of maximum batch size $B$ in the Poisson sampler: Taking $B$ to be large gets us closer to the privacy guarantee of the (untruncated) Poisson sampler, but increases the computation cost, since the batches are now larger (even if they contain dummy values with zero weight).
>
> > Line 254 -255, could the authors provide more details on selecting Ci in practice? And what is the computation complexity regarding this part?
>
> Indeed, while any choice of $C_i$’s are valid, there is an accuracy-vs-computation trade-off, in that, making $C_1$ smaller, $C_m$ larger and adding large number of intermediate $C_i$’s improves the accuracy of the lower bound at the cost of increased computational complexity.
> * As we note in the paper, we choose $C_1$ to be small enough, and $C_m$ to be large enough so that $P_{\cal S}(G_0)$ and $P_{\cal S}(G_{m+1})$ to be small enough. In particular, we chose the values to ensure $P_{\cal S}(G_0) + P_{\cal S}(G_{m+1}) \le e^{-40}$.
> * We chose $C_i$’s to be equally spaced in between $C_1$ and $C_m$ with a gap of $\Delta * \sigma^2$, where $\Delta$ is the desired discretization of the PLD. This is a heuristic choice guided by the intuition that we would like to discretize in buckets such that the privacy loss changes by $\Delta$, and since the privacy loss is approximately given as $\max_t x_t / \sigma^2$, the chosen gap means that this approximate privacy loss varies by $\Delta$ between buckets.
>
> These details are present in the implementation provided in the Colab link in the paper. But we will highlight these details more prominently in a revision.
>
> Please let us know if there are any further questions we can clarify, and if we have satisfactorily addressed the concerns of the reviewer, we kindly ask the reviewer to revisit the rating.

---

> > ### Comment · Reviewer_vHjL · 2024-08-11
> >
> > Thank the authors for the response and clarification. There are still some issues to be addressed:
> >
> > - W1 The authors highlight the difference between single-epoch and multi-epoch analysis of ABLQ. However, from my point of view, as reviewer k7t2 pointed out, the theoretical contribution of this part is limited.
> >
> > - W3 Thanks for the clarification. I wanted to emphasize that even with the new lower bounds for multi-epoch, the gap between Poisson subsampling and shuffle-based sampling is not addressed, especially the amplification effect in the low $\sigma$ region.
> >
> > - Q1 The statement in lines 47-49 is not supported by empirical results or further investigation, which may be inaccurate and misleading to other researchers/practitioners in the field. As far as I know, Opacus supports (distributed) Poisson sampling without the need to load the entire dataset (e.g., simply by indexing). I suggest the authors do their due diligence and check the implementation at https://opacus.ai/api/_modules/opacus/utils/uniform_sampler.html
> >
> > - Q2 The concern here is on the effect of batch size value used in averaging the sum of noisy gradients for privacy accounting. The batch size essentially reflects the sampling rate. However, truncated Poisson sampling distorts the probability of each sample being included in a batch. Has this been considered?
> >
> > - W2 \& Q3 Thank the authors for the response. I hope the authors will include these discussions in the manuscript.
> >
> > Given the amount of revisions required and the questions that remain, I am inclined to keep my initial assessment.

---

> ### Author Response · Authors · 2024-08-12
>
> > Opacus supports (distributed) Poisson sampling without the need to load the entire dataset (e.g., simply by indexing). I suggest the authors do their due diligence and check the implementation at https://opacus.ai/api/_modules/opacus/utils/uniform_sampler.html
>
> Thanks to the reviewer for the pointer. We apologize that our original text in lines 47-49 could be misinterpreted as “Opacus solution *requires* loading the dataset into the memory”. We meant to say that “loading into memory” is one way to facilitate this feature for small datasets. We will revise the text to make it more precise.
>
> To avoid any further misunderstanding between us and the reviewer on this matter, we summarize our view of the Opacus solution here: Opacus Poisson Subsampling works by using a unique identifier to index each example, and sampling the indices. They also provide a distributed sampler where each worker samples from a random subset of indices. This approach relies on a DataLoader that can provide efficient random access --- given an arbitrary index, read the corresponding data point efficiently. If random access is slow, the data pipeline will be slow, even though the (index) sampler is not the bottleneck.
>
> For small datasets, this can be easily supported by loading the data into memory. For large datasets that do not fit in the memory, various technical challenges need to be addressed to make it work. For example, many formats for raw data (e.g. `csv`, typically used for tabular or text data) or serialized data (e.g. `tfrecords` files, commonly used by the Tensorflow Datasets Catalog) do _not_ naturally support random indexing; moreover, random access can be much slower than sequential bulk reading in cases such as when the dataset is stored in a distributed file system. Various workarounds do exist depending on the resources and constraints, for example, by storing each example as an individual file in a fast SSD attached to the trainer machine. But those solutions usually need to be tailored to specific hardware and cluster infrastructures. On the contrary, we provide a **generic solution** that works for a large range of scales of dataset sizes and with **minimal assumption / requirement** on the underlying IO infrastructure.
>
> We agree that an empirical comparison of different technical solutions could be interesting but it is quite difficult to get right and useful as everyone has different configurations of (distributed) file systems, file format of preference, cluster configuration with communication channels of different properties, etc. Therefore it is quite far from the scope of our paper.
>
> Finally, regardless of the underlying technical solution, please note that ours is the _first large-scale_ DP training experiment with Poisson subsampling, comparing its feasibility with other solutions from multiple angles, including accounting, efficiency, and model utility.
>
> > truncated Poisson sampling distorts the probability of each sample being included in a batch
>
> It is true that due to truncation in Poisson subsampling, the probability that an example lands in a batch is reduced slightly. This is handled in the privacy analysis (Theorem 3.3).
>
> Please let us know if there are any further questions.

---

### Author Rebuttal · Authors · 2024-08-06

We thank all the reviewers for their thoughtful comments. In the attached pdf, we plot the noise multiplier $\sigma$ values against the number of epochs for different accounting methods, to answer a comment from Reviewer 9tRV.

---

### Decision · Program_Chairs · 2024-09-25

**Decision:**

Accept (poster)

**Comment:**

The main critiques of this paper were that the techniques may not be the most novel, and the contribution may be somewhat modest. Nonetheless reviewers agreed that this is an impactful advance on a very important problem, and thus worthy of acceptance.

In the discussion, Reviewers k7t2 and 9tRV brought up some points that they feel deserve mentioning in the final paper (highlighted in their reviews), and I anticipate the authors will take them into account.